# Occurrence of KPC-71 or the presence of NDM combined with a high expression of $bla_{SHV}$ contributes to low-level cefiderocol resistance in carbapenem-resistant *Klebsiella pneumoniae* in China

Hanxu Hong,[1,2,3] Yuxuan Liu,[1,2] Linping Fan,[1,4] Bowen Shi,[3] Ruihang Luo,[3] Chenye Xi,[5] Zhuoyuan Yang,[6] Wenjing Yang,[3] Xinyi Zeng,[3] DanDan Wei,[1] Yang Liu[1,2,7]

**ABSTRACT** Low-level cefiderocol (CFDC) resistance in carbapenem-resistant *Klebsiella pneumoniae* (CRKP) represents a growing clinical concern due to its potential for rapid resistance escalation and silent dissemination under antibiotic pressure. This study investigated the molecular epidemiology and resistance mechanisms of low-level CFDC-resistant CRKP in eastern China. Among 382 CRKP clinical isolates, 29 (7.6%) exhibited low-level CFDC resistance. Phylogenetic analysis categorized these isolates into three clusters: Clusters A and B comprised ST11-KL64 strains harboring $bla_{KPC}$ and virulence plasmids, whereas Cluster C primarily consisted of NDM-producing strains from diverse sequence types. Enzyme inhibition assays demonstrated that avibactam significantly reduced CFDC MICs (8-fold to 64-fold) in KPC-producing strains but had a limited impact (2-fold to 4-fold) in strains co-producing KPC and NDM or harboring the KPC-2 variant KPC-71. RT-qPCR showed that $bla_{SHV}$ was the only β-lactamase gene with significantly increased expression. Cloning experiments confirmed that NDM and SHV-12 contributed substantially to CFDC resistance, elevating MICs by 16-fold and 8-fold, respectively. Enzyme kinetics and molecular modeling indicated that KPC-71 enhances CFDC binding via a reduced $K_i$ and a potentially enlarged active-site pocket. These findings suggest that low-level CFDC resistance in CRKP is mainly driven by the presence of KPC-71 or NDM in combination with high $bla_{SHV}$ expression. In particular, a detailed analysis of KPC-71 revealed its critical role in diminishing CFDC susceptibility, offering important insights into the resistance mechanisms of emerging KPC variants.

**IMPORTANCE** Cefiderocol (CFDC) is a novel siderophore cephalosporin with potent activity against multidrug-resistant gram-negative bacteria, but it has not yet been approved for clinical use in China. The resistance mechanisms of CFDC among clinical isolates in China remain poorly understood, particularly in low-level resistant strains, which may pose a significant threat due to their potential for treatment failure and silent dissemination. In this study, we focused on the resistance mechanisms of low-level CFDC-resistant CRKP strains and revealed distinct resistance patterns that may require tailored treatment strategies. Notably, CFDC-avibactam combinations showed poor efficacy against strains producing NDM or novel KPC variants, whereas they may remain effective against strains producing KPC-2 with high $bla_{SHV}$ expression alone, resulting in ≥8-fold MIC reductions. Moreover, a detailed analysis of KPC-71 uncovered its critical role in reducing CFDC susceptibility, providing important insights into the resistance mechanisms of emerging KPC variants.

Address correspondence to Yang Liu, ly13767160474@sina.com.

The authors declare no conflict of interest.

See the funding table on p. 13.

KEYWORDS carbapenem-resistant *Klebsiella pneumoniae*, cefiderocol, low-level resistance, KPC-71, high expression of $bla_{SHV}$

The spread of carbapenemases has caused carbapenem-resistant *Klebsiella pneumoniae* (CRKP) to become a global health threat. CRKP is resistant to multiple commonly used antibiotics, presenting significant challenges to clinical treatment. Consequently, novel antimicrobial agents are urgently required to address this issue. Cefiderocol (CFDC), a novel siderophore cephalosporin, enters bacterial cells through catechol-type iron transporter receptors, such as the colicin I receptor (CirA) and the enterobactin receptor (FepA), to exert its antibacterial activity. *In vitro* studies have demonstrated CFDC's potent activity against various resistant bacteria (1, 2). Although CFDC has been approved by the European Medicines Agency and the Food and Drug Administration for treating infections caused by *K. pneumoniae*, it remains to be approved for use in China.

The Clinical and Laboratory Standards Institute (CLSI) defines CFDC resistance as a MIC >8 mg/L, whereas the European Committee on Antimicrobial Susceptibility Testing (EUCAST) adopts a lower breakpoint of MIC >2 mg/L. Consequently, isolates with MICs of 4–8 mg/L are considered resistant by EUCAST but remain susceptible under CLSI criteria. In this study, we refer to such isolates as low-level CFDC-resistant strains. Given that CLSI guidelines are predominantly used in China, these strains may be overlooked in clinical practice despite their potential to rapidly develop higher-level resistance under antibiotic pressure, leading to treatment failure and increased transmissibility (3, 4). Furthermore, low-level resistant strains pose a significant threat owing to their potential for cryptic transmission and the risk of large-scale dissemination following CFDC use, undermining the clinical efficacy of this promising antibiotic. Understanding CFDC resistance mechanisms in low-level resistant strains is crucial; however, current knowledge remains limited. Current studies mainly focus on genes related to iron transporter receptors, β-lactamases, and porins.

Mutations in the *cirA* gene cause CirA protein deficiency, contributing to CFDC-resistant strains (5–7). Our previous research revealed that most globally reported high-level CFDC-resistant CRKP strains (MIC >16 mg/L) exhibited CirA deficiency, which was not observed in low-level CFDC-resistant strains (8). Furthermore, NDM expression, combined with *cirA* mutations, can facilitate CFDC resistance development (9, 10). A prior study revealed the correlation between CFDC and ceftazidime-avibactam (CZA) resistance in *K. pneumoniae* strains carrying KPC (11), potentially due to the emergence of KPC variants; however, relevant research remains limited. Additionally, Ito et al. reported that the loss of OmpK35 or OmpK36 porins could contribute to CFDC resistance, resulting in a 2-fold to 4-fold increase in MIC (7). Some CFDC-resistant strains have also been reported to exhibit high virulence, which, combined with resistance, further complicates treatment strategies and presents new challenges for clinical therapy (12, 13).

In this study, we characterized 29 low-level CFDC-resistant CRKP strains selected from 390 clinical isolates to explore the molecular epidemiological characteristics and resistance mechanisms of low-level CFDC-resistant CRKP in Jiangxi, China. We demonstrated that the resistance mechanisms in these strains were associated with the presence of NDM, occurrence of the KPC-2 variant KPC-71, and high expression of $bla_{SHV}$, based on phenotypic-, genotypic-, and expression-level analyses. To our knowledge, we provide the first detailed evidence elucidating the relationship between KPC-71 and CFDC resistance. These findings offer a novel molecular perspective on the role of KPC variants in antibiotic resistance.

## MATERIALS AND METHODS

### Bacterial isolates

We collected 390 non-repetitive clinical CRKP isolates from a tertiary hospital in Jiangxi Province between January 2020 and January 2024. The isolates were identified by matrix-assisted laser desorption/ionization time-of-flight mass spectrometry (Bruker Daltonics). Carbapenem resistance was defined as resistance to imipenem or meropenem. Subsequently, only 382 isolates confirmed by PCR to carry at least one carbapenemase gene were included for further analysis, in order to ensure a focused investigation of carbapenemase-producing strains, which are the primary targets of CFDC.

### Antimicrobial susceptibility tests

Antimicrobial susceptibility testing (AST) was initially performed using a VITEK-2 system (bioMérieux, Lyon, France) in sentinel hospitals and validated using the broth microdilution method. The MIC of CFDC was determined using iron-depleted cation-adjusted Mueller–Hinton broth (ID-CAMHB), as described in EUCAST 2023 (https://www.eucast.org/). *Escherichia coli* ATCC 25922 was used as quality control strains for AST. The interpretative breakpoints were based on (14) M100 (14), whereas the breakpoints for the CFDC were based on EUCAST 2023. To evaluate the effect of β-lactamases on CFDC activity, CFDC MICs were determined with 4 mg/L avibactam (AVI) for inhibiting serine β-lactamases and 100 mg/L dipicolinic acid (DPA) for inhibiting metallo-β-lactamases.

### Whole-genome sequencing (WGS) and phylogenetic analysis

Genomic DNA from single clones was extracted using a TIANamp Bacterial DNA Kit (TianGen Biotech, Beijing, China). Genomes of the 29 CFDC-resistant strains were sequenced using the Illumina HiSeq platform (Illumina, San Diego, CA, USA). Automatic annotation of the genome sequences was performed using Prokka v1.14.646 (15). Sequence types (STs), capsular serotypes (KLs), and virulence scores were analyzed using Kleborate (16). Virulence genes, antibiotic resistance genes, and mobile genetic elements were predicted using the Virulence Factor Database (VFDB), CARD, and VRprofile, respectively. Putative virulence plasmids were inferred based on the presence of key virulence genes (*iucA*, *rmpA*, or *rmpA2*) and whole-genome alignment to the reference plasmid pLVPK (NC_005249.1) using fastANI v1.33 (17–19). A phylogenetic analysis of the 29 resistant isolates was conducted to investigate their evolutionary relationships. Roary v3.12 (20) was employed to perform a pan-genome analysis, identifying the core genes shared across the isolates. Multiple sequence alignments of the concatenated core genes were performed using MAFFT (21). Core-genome alignment was then used to construct a maximum-likelihood phylogeny with FastTree 2.1.11. The phylogenetic tree was visualized and further refined using tvBOT (22) to enhance clarity.

### RT-qPCR experiments

Ten CFDC-resistant and susceptible CRKP strains were randomly selected for real-time quantitative reverse transcription PCR (RT-qPCR) analysis of β-lactamase genes, following previously described methods (8). Gene expression levels were normalized to the average expression of the susceptible group strains, with 16sRNA expression used as the internal reference.

### Conjugation experiment

To evaluate the transferability of clinical plasmids carrying $bla_{KPC-2}$ and $bla_{KPC-71}$, conjugation experiments were performed as previously described (23). Briefly, azide-resistant *E. coli* J53 was used as the recipient strain, and *K. pneumoniae* KpCP182 (carrying $bla_{KPC-2}$) and Kp15 (carrying $bla_{KPC-71}$) were the donor strains. Transconjugants were selected on Luria–Bertani agar plates supplemented with 100 mg/L sodium azide and

16 mg/L ceftazidime. Conjugation efficiency was calculated by dividing the number of transconjugants by the number of recipient cells.

## Cloning and expression of β-lactamase genes

The sequences of $bla_{KPC-2}$ and $bla_{KPC-71}$ were amplified by PCR using primers that included homologous fragments of the pET28a vector (pKPC-F: TGCGGCCGCAAGCTTG TCGACTTACTGCCCGTTGACGCCC; pKPC-R: ATGGGTCGCGGATCCGAATTCATGTCACTGTATC GCCGTCTAGT). Other genes were amplified using different primers through the same amplification method. The PCR products were then ligated with linearized plasmids using restriction enzymes, via seamless cloning, and transformed into *E. coli* DH5α. Recombinant plasmids were screened by kanamycin (50 mg/L), PCR, and Sanger sequencing. Successful recombinant plasmids were transformed into *E. coli* BL21 (DE3). Gene expression was induced by adding 0.5 mM IPTG and incubating at 37°C.

## Protein purification and enzyme kinetics assay

KPC-2 and KPC-71 were expressed and purified as previously described (24). Briefly, the $bla_{KPC}$ genes were expressed in BL21 (DE3) cells harboring the constructed pET28a plasmids. Supernatants containing KPC proteins were collected and purified through nickel affinity chromatography. The concentration of purified KPC was determined by measuring absorbance at 280 nm using the Beer–Lambert law. Enzymatic kinetics were analyzed at room temperature using a spectrophotometer in PBS (pH 7.4). By monitoring the hydrolysis of different concentrations of antibiotics as substrates, the data were fitted to the Michaelis-Menten equation to calculate and analyze the kinetic parameters. When the substrate hydrolysis rate could not be saturated at measurable concentrations owing to a high $K_m$, the process curve was fitted to the equation $v = k_{cat}/K_m\ [E][S]$, where $[S] << K_m$, to determine the second-order rate constant $k_{cat}/K_m$ under steady-state conditions. To evaluate the affinity for KPC enzymes, fixed concentrations of KPC enzymes were incubated with various concentrations of FDC or AVI, followed by the addition of 100 µM nitrocefin ([S]). IC50 values were determined via nonlinear regression using GraphPad Prism 9.5.0, and $K_i$ values were calculated based on the Cheng-Prusoff equation (equation 1) (25).

$$K_i = \frac{IC_{50}}{1 + \frac{[S]}{K_m}} \tag{1}$$

## Molecular modeling

Modeller version 10.3 was used to generate a homology model of KPC-71, utilizing the KPC-2 structure (Protein Data Bank [PDB] ID: 2OV5) as the template. Molecular docking was performed using AutoDock (26), and the results were visualized with PyMOL.

## Statistical analysis

All statistical analyses were performed using GraphPad Prism 9.5.0. Categorical variables were analyzed using the $\chi^2$ or Fisher's exact test, whereas the Wilcoxon rank-sum test was used to compare resistance gene expression levels. $P < 0.05$ was considered statistically significant.

## RESULTS

### Clinical and genetic characteristics of low-level CFDC-resistant CRKP

Between 2020 and 2024, 382 CRKP isolates were collected at the First Affiliated Hospital of Nanchang University, of which 29 showed low-level CFDC resistance (MICs 4–8 mg/L), accounting for a resistance rate of 7.6%. No high-level CFDC-resistant strains were identified. These isolates exhibited extensive resistance to other antibiotics,

particularly cefoperazone-sulbactam, ceftazidime, and imipenem, with 100% resistance rates. Resistance rates < 50% were observed only for CZA, polymyxin, and tigecycline; however, these rates remained >20% (Fig. S1). WGS analysis revealed 27 distinct antibiotic resistance genes (ARGs) among the 29 isolates that conferred resistance to 11 classes of antibiotics (Fig. S2). Multidrug resistance significantly complicates treatment. Furthermore, WGS identified eight STs among the isolates, with ST11 being predominant (75.9%, 22/29), whereas ST1, ST15, ST29, ST290, ST307, ST1213, and ST5764 each accounted for 3.4% (1/29). Capsular serotypes (KLs) included six types, with KL64 most common (75.9%, 22/29), followed by KL19 and KL54 (6.9%, 2/29 each), and KL21, KL102, and KL103 (3.4% each, 1/29). Most isolates were derived from sputum and urine samples, with ST11 strains widely distributed across multiple hospital departments, whereas other STs were mainly confined to single departments (Fig. 1).

Regarding carbapenemase production, KPC was the predominant carbapenemase, present in 91.4% (349/382) of the strains, followed by NDM in 8.4% (32/382). Four strains carried both KPC and NDM. Among the 29 CFDC-resistant strains, KPC was present in 82.8% (24/29), and NDM in 27.6% (8/29), with three strains carrying both KPC and NDM. Notably, the rate of NDM carriage was significantly higher in CFDC-resistant strains than in all CRKP strains (27.6% vs 8.4%, $P < 0.001$), whereas the rate of KPC carriage was not significantly different (82.8% vs 91.4%, $P = 0.123$) (Fig. 2A).

## Phylogenetic analysis and profiles of CFDC resistance and virulence

Phylogenetic analysis based on WGS revealed that the CFDC-resistant isolates were divided into three clusters. Clusters A (21.4%) and B (51.7%) comprised ST11-KL64 isolates, whereas the remaining STs formed Cluster C (21.4%) (Fig. 1). In terms of CFDC resistance factors, most isolates in all three clusters carried the $bla_{SHV}$ gene (89.7%),

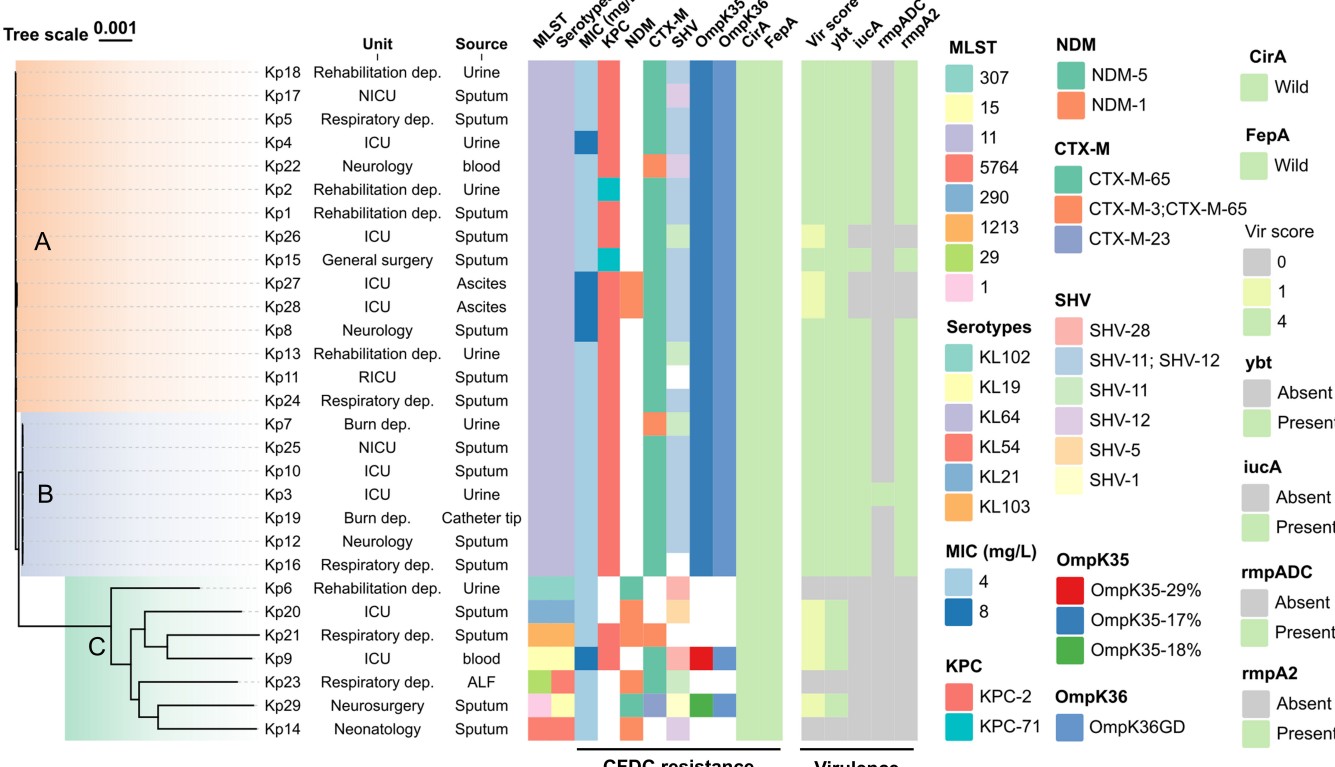

**FIG 1** Phylogenetic tree of CFDC-resistant strains. Leaf color blocks represent strains isolated from Cluster A (orange), Cluster B (blue), and Cluster C (cyan). The heatmap shows the following features (from left to right): MLST, Serotypes, CFDC MICs (mg/L), KPC, NDM, CTX-M, SHV, OmpK35 variant, OmpK36 variant, CirA, FepA, Virulence score, *ybt*, *iucA*, *rmpADC*, and *rmpA2*.

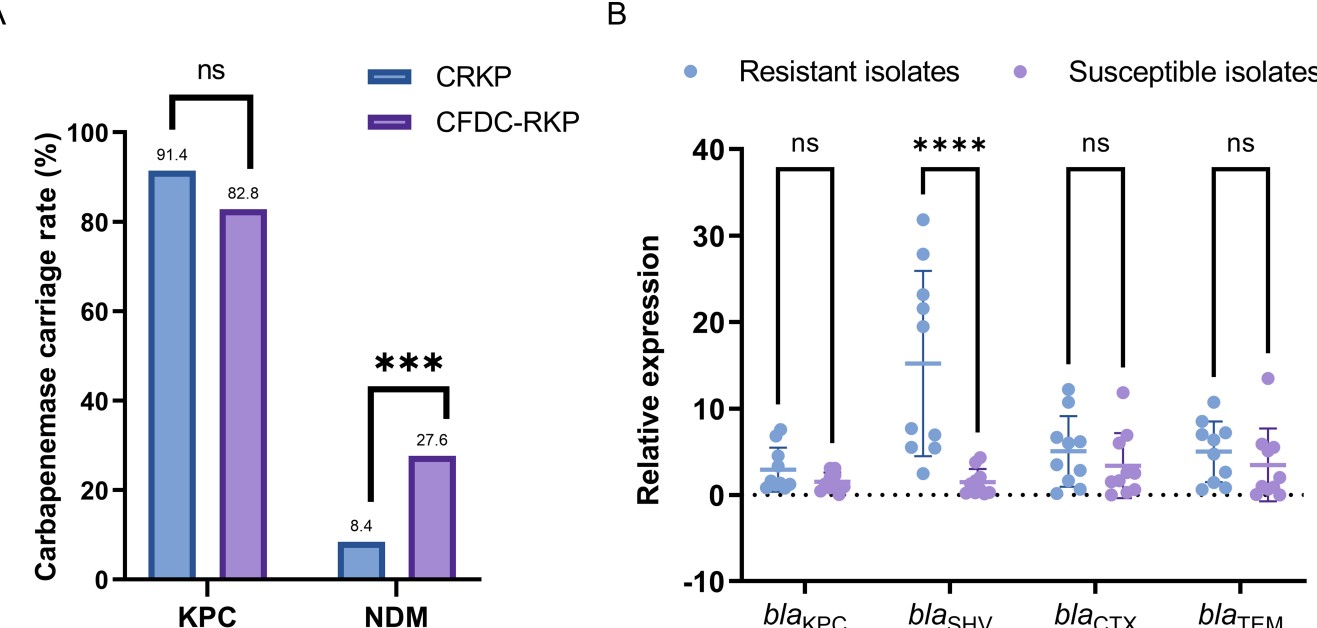

**FIG 2** The impact of β-lactamases on CFDC resistance. (A) Carriage rates of carbapenemases KPC and NDM in CRKP and CFDC-resistant strains. (B) Relative expression levels of β-lactamase genes in CFDC-resistant and susceptible strains (normalized to the mean expression level in susceptible strains).

whereas siderophore receptor genes associated with CFDC resistance (including *cirA* and *fiu*) were wild-type without mutations. Clusters A and B harbored $bla_{KPC}$ and $bla_{CTX}$-M$_{-M}$, along with OmpK35 truncations and OmpK36GD mutations, whereas cluster C predominantly carried $bla_{NDM}$ (85.7%), and only a small proportion of strains contained $bla_{KPC}$ and $bla_{CTX}$-M$_{-M}$. Based on virulence factor analysis, clusters A and B carried more virulence factors than cluster C, including *ybt* (100% vs 57.1%), *iucA* (86.4% vs 0%), *rmpADC* (4.5% vs 0%), and *rmpA2* (86.4% vs 0%). A comparison of genome-wide data with the classical virulence plasmid pLVPK revealed that 18 of the 21 strains in clusters A and B carried virulence plasmids, with only three lacking them. In contrast, none of the strains in cluster C carried virulence plasmids. Although both clusters A and B comprised ST11-KL64 strains, they were genetically different. Cluster B showed higher virulence scores and plasmid carriage, which may reflect genetic selection during strain evolution. Additionally, phylogenetic analysis based on PFGE yielded a clustering pattern similar to WGS, dividing the ST11-KL64 isolates into two closely related subgroups, whereas other ST types were more distantly related (Fig. S3). This indicates that PFGE may serve as a reliable tool for monitoring clonal strain transmission in hospitals.

### Effect of β-lactamases on CFDC resistance

To evaluate the contribution of β-lactamases to FDC resistance, we performed AVI and DPA inhibition experiments on CFDC-resistant strains. After the addition of the serine-β-lactamase (SBL) inhibitor AVI, the MICs of 19 out of 24 KPC-positive resistant strains decreased by ≥8-fold, with two strains showing a 64-fold decrease (to 0.0625 mg/L). Five strains showed a 2-fold to 4-fold decrease in MIC; three co-produced KPC and NDM, whereas the other two produced KPC-71 (Table 1). These results suggest that AVI significantly enhances CFDC activity against CRKP strains not producing metallo-β-lactamases (MBLs) but has limited efficacy against strains co-producing KPC and NDM or certain KPC variants. In contrast, the MBL inhibitor DPA reduced the CFDC MIC in NDM-positive strains by 2-fold to 8-fold (Table 1), but to a lesser extent than the reduction observed in KPC-positive strains. This suggests that the combination of DPA and CFDC has weaker activity against NDM-positive strains, possibly due to the co-production of other SBLs in these strains.

**TABLE 1** Antimicrobial activity of CFDC with or without β-lactamase inhibitors[a,d]

| Isolates | Carbapenemase | Other β-lactamases | MIC (mg/L) of CFDC | | | | |
|---|---|---|---|---|---|---|---|
| | | | Alone | +AVI | AVI fold[b] | +DPA[c] | DPA fold |
| Kp1 | KPC-2 | SHV-11; SHV-12; CTX-M-65; TEM-1 | 4 | 0.5 | 8 | | |
| Kp2 | KPC-71 | SHV-11; SHV-12; CTX-M-65; TEM-1 | 4 | 1 | 4 | | |
| Kp3 | KPC-2 | SHV-11; SHV-12; CTX-M-65; TEM-1 | 4 | 0.25 | 16 | | |
| Kp4 | KPC-2 | SHV-11; SHV-12; CTX-M-65; TEM-1 | 8 | 0.5 | 16 | | |
| Kp5 | KPC-2 | SHV-11; SHV-12; CTX-M-65; TEM-1 | 4 | 0.25 | 16 | | |
| Kp6 | NDM-5 | SHV-28 | 4 | 4 | 1 | 0.5 | 8 |
| Kp7 | KPC-2 | SHV-11; CTX-M-3; CTX-M-65; TEM-1 | 4 | 0.0625 | 64 | | |
| Kp8 | KPC-2 | SHV-11; SHV-12; CTX-M-65; TEM-1 | 8 | 1 | 8 | | |
| Kp9 | KPC-2 | SHV-28; CTX-M-65; TEM-1; OXA-1 | 8 | 0.5 | 16 | | |
| Kp10 | KPC-2 | SHV-11; SHV-12; CTX-M-65; TEM-1 | 4 | 0.5 | 8 | | |
| Kp11 | KPC-2 | CTX-M-65; TEM-1 | 4 | 0.0625 | 64 | | |
| Kp12 | KPC-2 | SHV-11; SHV-12; CTX-M-65; TEM-1 | 4 | 0.125 | 32 | | |
| Kp13 | KPC-2 | SHV-11; CTX-M-65; TEM-1 | 4 | 0.125 | 32 | | |
| Kp14 | NDM-1 | SHV-12 | 4 | 2 | 2 | 1 | 4 |
| Kp15 | KPC-71 | SHV-11; SHV-12; CTX-M-65; TEM-1 | 4 | 1 | 4 | | |
| Kp16 | KPC-2 | CTX-M-65; TEM-1 | 4 | 0.5 | 8 | | |
| Kp17 | KPC-2 | SHV-12: CTX-M-65; TEM-1 | 4 | 0.125 | 32 | | |
| Kp18 | KPC-2 | SHV-11; SHV-12; CTX-M-65; TEM-1 | 4 | 0.25 | 16 | | |
| Kp19 | KPC-2 | SHV-11; SHV-12; CTX-M-65; TEM-1 | 4 | 0.5 | 8 | | |
| Kp20 | NDM-1 | SHV-5 | 4 | 2 | 2 | 2 | 2 |
| Kp21 | KPC-2+NDM-1 | CTX-M-3; CTX-M-65; TEM-1; OXA-1 | 4 | 2 | 2 | 1 | 4 |
| Kp22 | KPC-2 | SHV-12; CTX-M-3; CTX-M-65; TEM-1 | 4 | 0.25 | 16 | | |
| Kp23 | NDM-1 | SHV-11; CTX-M-65; TEM-1 | 4 | 2 | 2 | 1 | 4 |
| Kp24 | KPC-2 | SHV-11; SHV-12; CTX-M-65; TEM-1 | 4 | 0.125 | 32 | | |
| Kp25 | KPC-2 | SHV-11; SHV-12; CTX-M-65; TEM-1 | 4 | 0.0625 | 64 | | |
| Kp26 | KPC-2 | SHV-11; CTX-M-65; TEM-1 | 4 | 0.25 | 16 | | |
| Kp27 | KPC-2+NDM-1 | SHV-11; SHV-12; CTX-M-65; TEM-1 | 8 | 2 | 4 | 1 | 8 |
| Kp28 | KPC-2+NDM-1 | SHV-11; SHV-12; CTX-M-65; TEM-1 | 8 | 2 | 4 | 1 | 8 |
| Kp29 | NDM-5 | SHV-1; CTX-M-65; TEM-1; OXA-1 | 4 | 4 | 1 | 1 | 4 |

[a]AVI, avibactam; CFDC, cefiderocol; DPA, dipicolinic acid; MIC, minimum inhibitory concentration.
[b]AVI fold or DPA fold refers to the fold reduction in the MIC of CFDC after the addition of AVI or DPA.
[c]DPA inhibition tests were only conducted on isolates carrying metallo-β-lactamases.
[d]Empty cells indicate that the corresponding isolates do not produce metallo-β-lactamases; therefore, DPA inhibition or combined DPA + AVI inhibition tests were not performed.

Additionally, to identify the primary SBLs responsible for CFDC resistance in low-level resistant strains, we randomly tested the expression of various β-lactamases in ten resistant and susceptible strains. As shown in Fig. 2B, the expression of the $bla_{SHV}$ gene was significantly higher in low-level resistant strains ($P < 0.005$), whereas the expression of other genes, although slightly elevated compared with that in the susceptible strains, showed no significant differences. As shown in Fig. 1, the 29 CFDC-resistant strains carried multiple $bla_{SHV}$ genes, including $bla_{SHV-1}$, $bla_{SHV-5}$, $bla_{SHV-11}$, $bla_{SHV-12}$, and $bla_{SHV-28}$. Among them, co-expression of $bla_{SHV-11}$ and $bla_{SHV-12}$ was common in ST11 strains (68.2%), which may partially contribute to the elevated $bla_{SHV}$ expression observed in resistant strains. Notably, we found that Kp22, despite carrying only $bla_{SHV-12}$, exhibited a high level of $bla_{SHV}$ expression. Furthermore, long-read sequencing revealed that $bla_{SHV-12}$ was present on both the chromosome and plasmid. We compared the genetic contexts of the two loci and found that the chromosomal $bla_{SHV}$ collinear block exhibited partial structural conservation with the IS26-associated plasmidic $bla_{SHV}$ region (83.4% coverage and 99% nucleotide identity over 5,257 bp; Fig. S4). This finding underscores the necessity of RT-qPCR for monitoring $bla_{SHV}$ expression levels and highlights the importance of combining RT-qPCR with long-read sequencing for an accurate assessment of $bla_{SHV}$ gene copies.

Through cloning experiments, we further elucidated the independent contributions of different β-lactamase genes to CFDC resistance. After introducing NDM-1 and NDM-5 into *E. coli* BL21(DE3), CFDC MIC increased 16-fold (to 1 mg/L), whereas KPC-2 only led to a 2-fold increase (to 0.125 mg/L) (Table 2). Among the $bla_{SHV}$ genes, SHV-12 exhibited the most significant impact on CFDC MIC, increasing it 8-fold (to 0.5 mg/L), followed by SHV-5 (4-fold, to 0.25 mg/L). In contrast, SHV-1, SHV-11, and SHV-28 had relatively weaker effects, each causing a 2-fold increase (to 0.125 mg/L). In summary, CFDC resistance in NDM-producing strains is primarily attributed to NDM, whereas in KPC-producing strains, high expression of $bla_{SHV}$—particularly $bla_{SHV-12}$—combined with the presence of other serine β-lactamases plays a key role. Notably, AVI significantly reduces CFDC resistance in KPC-producing strains lacking metallo-β-lactamases, demonstrating strong *in vitro* antibacterial activity when combined with CFDC.

## Thr180_Ser181insSer variant of KPC-2 decreases CFDC susceptibility

During the analysis of resistance mechanisms, we identified two low-level CFDC-resistant strains, Kp15 and Kp2, both producing KPC-71. Genetic analysis revealed that $bla_{KPC-71}$ is a variant of $bla_{KPC-2}$, with a three-nucleotide insertion (TCA) between positions 540 and 541, leading to the insertion of a serine residue between positions 180 and 181 in the amino acid sequence (Fig. 3A). To evaluate the impact of KPC-71 on resistance, we isolated KpCP182, a KPC-2-producing strain homologous to Kp15, from the same patient 18 days earlier and confirmed its homology using PFGE (Fig. S5). Antibiotic susceptibility testing revealed that compared with KpCP182, Kp15 exhibited an 8-fold increase in the MIC of CFDC and a 32-fold increase in that of CZA. In contrast, the MICs of meropenem and imipenem decreased by 32-fold and 16-fold, respectively, although susceptibility to both remained unrestored (Table 3). Comparative plasmid analysis revealed that the KPC plasmids in strains Kp15 and KpCP182 were nearly identical except for the $bla_{KPC}$ gene (Fig. S6). The genetic environment surrounding $bla_{KPC-2}$ and $bla_{KPC-71}$ was identical and characterized by the structure Tn1721-IS*Kpn6*-$bla_{KPC}$-IS*Kpn27*-Tn3 (Fig. S7). Conjugation experiments demonstrated that the transconjugant frequencies of Kp15-pKPC-71 and KpCP182-pKPC-2 were $(6.46 \pm 0.37) \times 10^{-6}$ and $(5.18 \pm 0.59) \times 10^{-6}$, respectively. The above results indicate that Kp15-pKPC-71 possesses the ability for conjugative transfer, and mutation of the $bla_{KPC}$ gene did not affect the conjugation efficiency of the plasmid.

To further validate the effect of KPC-71, we cloned and expressed $bla_{KPC-71}$ and $bla_{KPC-2}$ in the same genetic background to examine their effects on antibiotic resistance. The results demonstrated that the CFDC MIC of *E. coli* BL21DE3/pKPC-71 expressing $bla_{KPC-71}$ was 16-fold higher than that of the vector control strain *E. coli* BL21DE3/pET28a and 8-fold higher than that of *E. coli* BL21DE3/pKPC-2 expressing $bla_{KPC-2}$, confirming

TABLE 2 Antibiotic susceptibility of *E. coli* BL21(DE3) isolates expressing β-lactamases[a]

| Strain | Phenotype | MIC (mg/L) | | MIC fold change[b] | |
|---|---|---|---|---|---|
| | | CFDC | MEM | CFDC | MEM |
| BL21DE3 | Wild type | 0.06 | 0.06 | –[c] | – |
| NDM-1 | Carbapenemase | 1 | 4 | 16 | 64 |
| NDM-5 | Carbapenemase | 1 | 4 | 16 | 64 |
| KPC-2 | Carbapenemase | 0.125 | 2 | 2 | 32 |
| SHV-1 | GSBL | 0.125 | 0.06 | 2 | 1 |
| SHV-5 | ESBL | 0.25 | 0.06 | 4 | 1 |
| SHV-11 | GSBL | 0.125 | 0.06 | 2 | 1 |
| SHV-12 | ESBL | 0.5 | 0.06 | 8 | 1 |
| SHV-28 | GSBL | 0.125 | 0.06 | 2 | 1 |

[a]CFDC, cefiderocol; ESBL, extended-spectrum β-lactamases; GSBL, general-spectrum β-lactamases; MEM, meropenem; MIC, minimum inhibitory concentration. *E. coli* BL21DE3 and its isogenic mutants must be induced with 0.5 mM IPTG both before and during MIC testing to ensure the normal expression of the target gene on the pET28a plasmid.
[b]MIC fold change refers to the fold change in MIC of the mutant compared to the parent strain.
[c]"–" indicates that MIC fold change values are not applicable for the reference strain.

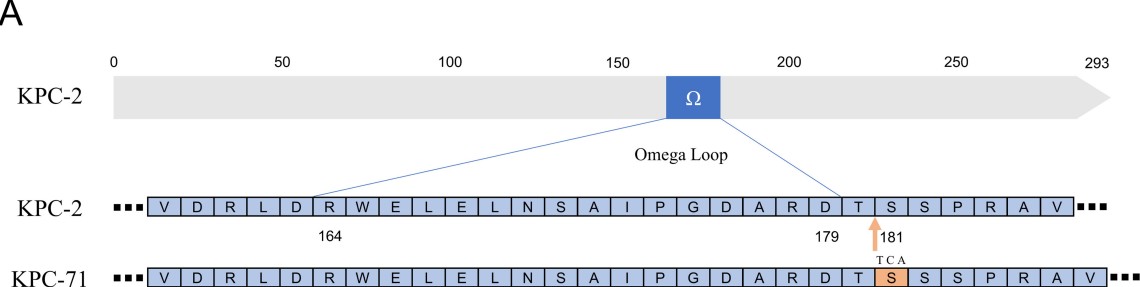

FIG 3  KPC-2 and KPC-71 amino acid sequence, three-dimensional protein structure comparison, and binding mode of KPC-71 with CFDC. (A) The amino acid sequence comparison of KPC-2 and KPC-71. The Ω-loop region (positions 164–179) is highlighted, with the inserted serine (Ser) residue shown in yellow. (B, C) The three-dimensional structures of KPC-2 (green) and KPC-71 (cyan) are compared. The color coding is as follows: orange, Ser-70; wheat, Ω-loop; purple, (Continued on next page)

Fig 3 (Continued)

KPC-2 sequence at positions 180-181; red, KPC-71 sequence at positions 180-182. (D) Molecular docking model of KPC-71 with CFDC. The magnified view shows hydrogen bond interactions between CFDC and the active site of KPC-71, involving key amino acid residues such as SER-70, SER-130, ASN-132, ASN-170, THR-237, HIS-274, and GLU-276.

that KPC-71 significantly decreased susceptibility to CFDC (Table 3). Moreover, compared with *E. coli* BL21DE3/pKPC-2, *E. coli* BL21DE3/pKPC-71 showed a 64-fold increase in the CZA MIC and restored susceptibility to imipenem and meropenem.

## Enzyme kinetics and molecular modeling of KPC-71 and KPC-2

We further measured the enzyme kinetics of KPC-2 and KPC-71. Because of the low initial hydrolytic activities of KPC-2 and KPC-71 against CFDC, the hydrolytic efficiency of CFDC could not be determined. To evaluate the effect of KPC-71 on CFDC, we used CFDC as an inhibitor. The results showed that the $Ki_i$ value of CFDC for KPC-2 was 38 times that of KPC-71, indicating that KPC-2 had a significantly lower affinity for CFDC than KPC-71 (Table 4). Furthermore, KPC-2 demonstrated considerable hydrolytic activity against carbapenems, such as meropenem and imipenem, whereas the hydrolytic activity of KPC-71 against these antibiotics was challenging to measure. Meanwhile, compared with wild-type KPC-2, the $K_i$ value of inhibitor AVI for KPC-71 was 12 times higher, suggesting a reduced affinity and weakened inhibitory effect of AVI on KPC-71.

To investigate the atomic-level impact of the Thr180_Ser181insSer mutation on KPC, we constructed three-dimensional structural models of KPC-2 and KPC-71 and simulated their binding modes with CFDC. Previous studies have indicated that the core of the active site of KPC contains serine at position 70, surrounded by the omega-loop (Arg164 to Asp179) (27). Structural comparisons reveal that the Thr180_Ser181insSer mutation causes an outward expansion of the Ω-loop and its adjacent regions, resulting in a larger binding pocket compared to KPC-2 (Fig. 3B and C). This allows better accommodation of CFDC and CAZ, which have larger R2 side chains, leading to a higher affinity. Conversely, the simpler and smaller molecule AVI exhibits reduced affinity, explaining the observed increase in MIC for CFDC and CAZ and the significant increase in MIC for CZA. Molecular docking analysis further supports this hypothesis, showing that CFDC forms multiple hydrogen bonds with key active-site residues, including Ser70, Ser130, Asn132, Asn170, Thr237, His274, and Glu276 (Fig. 3D). The outward shift of the Ω-loop reduces steric hindrance between Asn170 and Asn132, facilitating the binding of CFDC, which contains a bulkier catechol-substituted R2 side chain. These interactions contribute to the stable binding of CFDC within KPC-71, resulting in a higher affinity than in KPC-2. Additionally, binding free energy calculations show that KPC-71 has a negative binding free energy value of −8.2 kcal/mol for CFDC, further confirming the favorable interaction between KPC-71 and CFDC.

## DISCUSSION

This study demonstrated that in the Jiangxi region of eastern China, CFDC-resistant CRKP strains primarily exhibited low-level resistance (7.4% resistance rate), with no high-level

TABLE 3 Antibiotic susceptibility of the strains used in this study (mg/L)[a]

| Strain | KPC subtype | MIC (mg/L) | | | | |
|---|---|---|---|---|---|---|
| | | CFDC | CAZ | CZA | MEM | IPM |
| *K. pneumoniae* KpCP182 | KPC-2 | 1 | 256 | 4 | 64 | 64 |
| *K. pneumoniae* Kp15 | KPC-71 | 4 | 256 | 128 | 2 | 4 |
| *E. coli* BL21DE3 (pET28a) | –[b] | 0.06 | 0.5 | 0.125 | <0.5 | <0.5 |
| *E. coli* BL21DE3 (pET28a-KPC-2) | KPC-2 | 0.125 | 32 | 0.25 | 32 | 64 |
| *E. coli* BL21DE3 (pET28a-KPC-71) | KPC-71 | 1 | 64 | 16 | <0.5 | <0.5 |

[a]CAZ, ceftazidime; CZA, ceftazidime-avibactam; IPM, Imipenem.
[b]"–" indicates that the strain does not carry a KPC subtype.

**TABLE 4** Kinetic parameters of the β-lactamases KPC-2 and KPC-71[a]

| Antibiotic | KPC-2 | | | | KPC-71 | | | |
|---|---|---|---|---|---|---|---|---|
| | $K_m$ | $k_{cat}$ | $k_{cat}/K_m$ | $K_i$ | $K_m$ | $k_{cat}$ | $k_{cat}/K_m$ | $K_i$ |
| Nitrocefin | 7.190 | 23.577 | 3.279 | –[b] | 8.829 | 16.617 | 1.882↓ | – |
| Cefiderocol | ND | ND | ND | 11.920 | ND[c] | ND | ND | 0.315↓ |
| Ceftazidime | ND | ND | 0.007 | – | ND | ND | 0.004↓ | – |
| Meropenem | 13.582 | 3.923 | 0.289 | – | ND | ND | ND↓ | – |
| Imipenem | 37.350 | 12.902 | 0.345 | – | ND | ND | ND↓ | – |
| Avibactam | – | – | – | 0.014 | – | – | – | 0.168↑ |

[a]Arrows in the table indicate the changing trend of KPC-71 kinetic parameters compared with corresponding KPC-2 kinetic parameters. ND, not determined due to a low initial rate of hydrolysis. $k_{cat}$, turnover; $K_m$, Michaelis constant (affinity); $k_{cat}/K_m$, specificity constant (hydrolysis); $K_i$, inhibition constant (affinity).
[b]"–" indicates that the value was not determined.
[c]ND, not determined due to a low initial rate of hydrolysis.

resistant strains identified. A recent meta-analysis of CFDC resistance (28), based on EUCAST breakpoint thresholds, showed a global resistance rate of 12.4% for carbapenem-resistant *Enterobacteriaceae*, which is higher than that observed in the present study. The resistance rate of NDM-producing *Enterobacteriaceae* was 38.8%, consistent with our finding of higher NDM prevalence in CFDC-resistant strains. This aligns with reports that NDM enzymes confer higher resistance to CFDC than other carbapenemases (29). Although previous studies showed that dipicolinic acid (DPA) could restore carbapenem susceptibility by inhibiting NDM (30), our results suggest DPA has a limited effect on reversing CFDC resistance in NDM producers, warranting further investigation. Although previous studies showed that dipicolinic acid (DPA) could restore carbapenem susceptibility by inhibiting NDM, our results suggest DPA has a limited effect on reversing CFDC resistance in NDM producers, warranting further investigation. Furthermore, although these low-level resistant strains had limited resistance to CFDC, they demonstrated broad resistance to other commonly used antibiotics, particularly 100% resistance to cefoperazone-sulbactam, ceftazidime, and imipenem. This highlights the limited therapeutic options available for treating these strains, which further complicates their clinical management.

Phylogenetic analysis revealed significant genetic differences among CRKP strains, particularly regarding β-lactamase diversity and virulence factor carriage. Compared with Cluster C strains, which lack virulence plasmids, Clusters A and B, comprising ST11-KL64 strains, exhibited significantly higher rates of virulence genes and plasmid carriage, with Cluster B showing the highest virulence levels. Cao et al. suggested that CR-hvKP may possess stronger resistance due to the production of additional iron carriers, although this difference was not statistically significant (31). China is the earliest reported region for CR-hvKP and remains a major hotspot for its prevalence. Among these, ST11-KL64 is the predominant clonal type, which is already distributed globally and frequently carries the $bla_{KPC-2}$ resistance and virulence plasmids with partial deletions of virulence factors (32–34). These findings highlight the importance of monitoring ST11-KL64 CR-hvKP strains, focusing on their CFDC resistance and transmission dynamics. Antibiotic susceptibility testing combined with PFGE may be a feasible approach for real-time surveillance.

The impact of AVI in enzyme inhibition experiments suggests that the combination of CFDC and AVI could be a promising therapeutic strategy for treating low-level CFDC-resistant CRKP strains that do not produce metallo-β-lactamases. Among the 19 KPC-2-producing strains that do not produce metallo-β-lactamases, a significant MIC reduction of ≥8-fold was observed in all strains after the addition of AVI, with some showing a 64-fold decrease. This confirms the contribution of serine-type β-lactamases to low-level CFDC resistance in CRKP strains. Further β-lactamase expression analyses revealed that $bla_{SHV}$ was the only β-lactamase gene significantly overexpressed in resistant strains. Cloning experiments demonstrated that among the five $bla_{SHV}$ variants detected, $bla_{SHV-12}$ had the most pronounced impact on CFDC resistance.

These findings suggest that high expression of $bla_{SHV}$, particularly $bla_{SHV-12}$, is likely a major factor contributing to the low-level resistance of CFDC-resistant strains in Jiangxi. Previous studies have shown that most *K. pneumoniae* strains harbor chromosomal $bla_{SHV}$ variants, such as $bla_{SHV-1}$ and $bla_{SHV-11}$, which are considered evolutionary ancestors of plasmid-borne variants like $bla_{SHV-5}$ and $bla_{SHV-12}$ (35, 36). This may explain the co-existence of $bla_{SHV-11}$ and $bla_{SHV-12}$ in our ST11 low-level resistant strains. A recent finding suggests that high copy numbers of $bla_{SHV}$ lead to elevated expression, thereby causing high-level resistance to CFDC (37). Additionally, tandem amplification of β-lactamase genes reduces the effectiveness of antibiotics (38, 39). In our study, strain Kp22 revealed an alternative mechanism for high expression: $bla_{SHV-12}$ was present on both the plasmid and chromosome, enabling dual expression.

For KPC-71, we focused on analyzing a strain, Kp15, which evolved from a KPC-2-producing homologous strain to a KPC-71 producer. The Thr180_Ser181insSer mutation altered the *in vitro* susceptibility to CFDC and CZA, converting it from susceptible to resistant, while enhancing susceptibility to meropenem and imipenem. However, MIC values remained above the CLSI breakpoint for susceptibility. These results were confirmed through cloning and expression experiments. Notably, plasmid pKPC-71 in Kp15 demonstrated conjugative transferability, emphasizing the importance of monitoring the dissemination of KPC-71.

Previous reports have suggested that increased resistance to CFDC might be attributed to the binding of KPC variants to CFDC rather than hydrolysis (40, 41). Our enzyme kinetics experiments further confirmed this, showing that compared with KPC-2, KPC-71 exhibited a significantly higher affinity for CFDC but a reduced affinity for AVI. From an antibiotic structural perspective, CFDC is similar to ceftazidime, differing only in the presence of a catechol group on the C-3 side chain. Both antibiotics share a larger R2 side chain than cefotaxime and cefalexin (42, 43). The insertion of serine in KPC-71 enlarges the cefem-binding pocket composed of the omega-loop and amino acid residues surrounding Ser-70, thereby increasing the substrate-binding space and affinity for larger antibiotics. Through 3D structural modeling, we further demonstrated how the KPC-71 mutation altered its binding pocket, affecting antibiotic affinity. This discovery provides new insights into the role of KPC variants in resistance mechanisms and offers important clues for future investigations into resistance mechanisms of similar variants, such as KPC-114, KPC-123, and KPC-108 (44). It should be noted that there is currently no rapid and accurate molecular detection method for these variants. Therefore, the development and optimization of molecular screening techniques for such variants are urgently needed.

This study had some limitations. First, our study focused only on the CFDC resistance mechanisms in Jiangxi, limiting the scope of the findings. To expand the applicability and representativeness of the results, further studies across a broader geographic region and larger sample sizes are needed to reveal the regional differences and epidemiological features of CFDC resistance. Second, we were unable to determine the complete and individual resistance mechanisms of the 29 low-level CFDC-resistant CRKP isolates. Third, the explanation of why DPA failed to significantly reduce the MIC of CFDC in these resistant strains remains unclear, but it may be related to SHV or CTX coproduction in NDM-producing strains; this requires further validation.

## Conclusions

This study provides an in-depth analysis of the molecular epidemiology and resistance mechanisms of low-level CFDC-resistant CRKP strains in Jiangxi, revealing that the presence of NDM or the emergence of KPC-71, accompanied by the high expression of $bla_{SHV}$, is a key factor driving low-level CFDC resistance. A detailed analysis of KPC-71 further elucidated its significant role in the marked increase in CFDC and CZA MICs and the underlying mechanisms. These findings not only broaden our understanding of KPC variants' roles in antibiotic resistance but also highlight the potential synergistic

impact of these β-lactamases on CFDC resistance, providing valuable insights into the development of molecular diagnostic tools and targeted intervention strategies.

## ACKNOWLEDGMENTS

This work was supported by the National Natural Science Foundation of China (grant number 82102411, 82260403, 32370195) and the Clinical Research Nurture Project of the First Affiliated Hospital of Nanchang University (grant number YFYLCYJPY202201).

## AUTHOR AFFILIATIONS

[1]Department of Clinical Laboratory, The First Affiliated Hospital, Jiangxi Medical College, Nanchang University, Nanchang, China
[2]School of Public Health, Jiangxi Medical College, Nanchang University, Nanchang, China
[3]First Clinical Medical College, Nanchang University, Nanchang, China
[4]Center for Molecular Diagnosis and Precision Medicine, The First Affiliated Hospital, Jiangxi Medical College, Nanchang University, Nanchang, China
[5]College of Stomatology, Nanchang University, Nanchang, China
[6]Second Clinical Medical College, Nanchang University, Nanchang, China
[7]Jiangxi Medical Center for Critical Public Health Events, The First Affiliated Hospital, Jiangxi Medical College, Nanchang University, Nanchang, China

## AUTHOR ORCIDs

Hanxu Hong  http://orcid.org/0009-0005-0937-429X
Yuxuan Liu  http://orcid.org/0000-0003-1239-0539
Yang Liu  http://orcid.org/0009-0009-1869-3558

## FUNDING

| Funder | Grant(s) | Author(s) |
| --- | --- | --- |
| National Natural Science Foundation of China | 32370195 | Yang Liu |
| National Natural Science Foundation of China | 82102411 | Yang Liu |
| National Natural Science Foundation of China | 82260403 | Yang Liu |

## AUTHOR CONTRIBUTIONS

Hanxu Hong, Investigation, Visualization | Yuxuan Liu, Methodology, Validation | Linping Fan, Validation | Bowen Shi, Conceptualization, Validation | Ruihang Luo, Validation | Chenye Xi, Visualization | Zhuoyuan Yang, Validation | Wenjing Yang, Investigation, Validation | Xinyi Zeng, Data curation, Project administration | DanDan Wei, Conceptualization, Supervision | Yang Liu, Formal analysis, Project administration, Resources, Supervision

## DATA AVAILABILITY

Complete sequences of the 29 CFDC low-level resistant CRKP isolates have been deposited on NCBI with accession numbers SAMN45153714, SAMN44805724, SAMN45153780, SAMN45153812, SAMN45153892, SAMN45153950, SAMN45153955–SAMN45153958, SAMN45153961, SAMN45153974, SAMN45154023, SAMN45154614, SAMN45176933, SAMN42768826, SAMN45194979, SAMN45194984–SAMN45194986, SAMN45194990, SAMN45195111, SAMN45174027, SAMN45174110, and SAMN45176168, SAMN45176169.

## ETHICS APPROVAL

Approval was obtained from the ethics committee of the First Affiliated Hospital of Nanchang University [approval/reference number: (2023)CDYFYYLK(01-062)].

## ADDITIONAL FILES

The following material is available online.

### Supplemental Material

**Supplemental figures (Spectrum01593-25-s0001.docx).** Fig. S1 to S7.

### Open Peer Review

**PEER REVIEW HISTORY (review-history.pdf).** An accounting of the reviewer comments and feedback.

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
