## [Reviewer comments · Microbiology Spectrum]

Microbiology Spectrum

Occurrence of KPC-71 or presence of NDM combined with high expression of blaSHV contributes to low-level cefiderocol resistance in carbapenem-resistant *Klebsiella pneumoniae* in China

Hanxu Hong, Yuxuan Liu, Lin-Ping FAN, Bowen Shi, Ruihang Luo, Chenye Xi, Zhuoyuan Yang, WenJing Yang, Xinyi Zeng, Dan Wei, and Yang Liu

Corresponding Author(s): Yang Liu, First Affiliated Hospital of Nanchang University

Review Timeline:

Submission Date:	June 10, 2025
Editorial Decision:	June 16, 2025
Revision Received:	June 22, 2025
Accepted:	June 24, 2025

Editor: Gabriele Arcari

Reviewer(s): The reviewers have opted to remain anonymous.

Transaction Report:

DOI: <https://doi.org/10.1128/spectrum.01593-25>

Re: Spectrum01593-25 (Occurrence of KPC-71 or presence of NDM combined with high expression of blaSHV contributes to low-level cefiderocol resistance in carbapenem-resistant *Klebsiella pneumoniae* in China)

Dear Mr. Yang Liu:

Thank you for the privilege of reviewing your work. Below you will find my comments, instructions from the Spectrum editorial office, and the reviewer comments.

I am pleased to inform you that your manuscript has been editorially accepted for publication. However, there are a few additional questions in the submission form that need to be answered before the final decision. Once these are completed, please return your submission so that I can move your paper forward to acceptance.

Revision Guidelines

Sincerely,
Gabriele Arcari
Editor
Microbiology Spectrum

Re: Spectrum01593-25R1 (Occurrence of KPC-71 or presence of NDM combined with high expression of blaSHV contributes to low-level cefiderocol resistance in carbapenem-resistant *Klebsiella pneumoniae* in China)

Dear Mr. Yang Liu:

Your manuscript has been accepted, and I am forwarding it to the ASM production staff for publication. Your paper will first be checked to make sure all elements meet the technical requirements. ASM staff will contact you if anything needs to be revised before copyediting and production can begin. Otherwise, you will be notified when your proofs are ready to be viewed.

Sincerely,
Gabriele Arcari
Editor
Microbiology Spectrum